# Quercetin as an Auxiliary Endodontic Irrigant for Root Canal Treatment: Anti-Biofilm and Dentin Collagen-Stabilizing Effects In Vitro

**DOI:** 10.3390/ma14051178

**Published:** 2021-03-03

**Authors:** Zhuo Liu, Xiangli Feng, Xiangyao Wang, Shiyuan Yang, Jing Mao, Shiqiang Gong

**Affiliations:** 1Center of Stomatology, Tongji Hospital, Tongji Medical College, Huazhong University of Science and Technology, Wuhan 430030, China; liuzhuo@hust.edu.cn (Z.L.); m202076256@hust.edu.cn (X.W.); yangshiyuan1996@sjtu.edu.cn (S.Y.); 2Hubei Province Key Laboratory of Oral and Maxillofacial Development and Regeneration, Wuhan 430022, China; 3Department of Stomatology, Hubei Provincial Hospital of TCM, Wuhan 430061, China; doctorfengxiangli@163.com

**Keywords:** antibacterial activity, endodontic irrigant, enterococcus faecalis, quercetin, root canal

## Abstract

Bacterial reinfection and root fracture are the main culprits related to root canal treatment failure. This study aimed to assess the utility of quercetin solution as an adjunctive endodontic irrigant that does not weaken root canal dentin with commitment anti-biofilm activity and bio-safety. Based on a noninvasive dentin infection model, dentin tubules infected with *Enterococcus faecalis* (*E. faecalis*) were irrigated with sterile water (control group), and 0, 1, 2, 4 wt% quercetin-containing ethanol solutions. Live and dead bacteria percentages in *E. faecalis* biofilms were analyzed by confocal laser scanning microscopy (CLSM). Elastic modulus, hydroxyproline release and X-ray photoelectron spectroscopy (XPS) characterization were tested to evaluate the irrigants’ collagen-stabilizing effect. The cytotoxicity was tested by CCK-8 assay. Quercetin increased the proportion of dead bacteria volumes within *E. faecalis* and improved the flexural strength of dentin compared to control group (*p* < 0.05). Quercetin-treated dentin matrix had less elasticity loss and hydroxyproline release after collagenase degradation (*p* < 0.05). Moreover, quercetin solutions revealed an increase in the C-O peak area under both C1s and O1s narrow-scan spectra of XPS characterization, and no cytotoxicity (*p* > 0.05). Quercetin exhibited anti-biofilm activity, a collagen-stabilizing effect with cytocompatibility, supporting quercetin as a potential candidate for endodontic irrigant.

## 1. Introduction

Root canal treatment is the method of choice for treating oral diseases associated with pulp and periapical infection. Endodontic microsurgery is often the last option to resolve persistent lesions with etiology related to complex root canal anatomy when non-surgical treatment is not effective [1]. The success of these endodontic treatments mainly relies on the thorough chemo-mechanical debridement of the infected root canals through the effective use of disinfectants and instrumentation, followed by sealing the enlarged, disinfected canals with appropriate root filling materials [2]. However, the bacterial elimination efficacy of root canal treatment is often compromised due to various reasons, such as untouched residual debris, and incomplete root canal filling that would allow secondary bacterial colonization [3,4,5]. In an ocean of failure cases, *Enterococcus faecalis* (*E. faecalis*) has been reported to present in the root canal as a biofilm alone or as the predominant bacterial specie [6,7,8]. In this regard, besides meticulous instrumentation, auxiliary endodontic medicaments (e.g., calcium hydroxide) and irrigants (e.g., sodium hypochlorite) with antibacterial activity are adopted by clinicians during root canal treatment, in order to disinfect the canal system and prevent reinfection. However, the risk of using these disinfectants is that they will cause changes in the chemical composition of the dentin (e.g., the biodegradation of demineralized dentin collagen) as long as disinfectants are applied in high concentrations for long periods [9,10]. In turn, these changes reduced the biomechanical properties of dentin, such as microhardness [11,12,13], flexural strength [14,15], modulus of elasticity [15,16], and the fracture resistance under normal masticatory loads. This is considered to be another risk factor for the root canal treatment failure. Therefore, endodontic disinfectants that kill *E. faecalis* whilst not compromising the biomechanical properties of dentin are in great demand [17,18,19,20].

Quercetin (Figure 1) is a natural bioflavonoid found in abundance in food and beverages such as broccoli, onions, tea, apples, and berries. It has many biological and pharmacological benefits, such as anti-inflammatory, anti-allergic and antioxidant properties, owing to its ability to remove reactive oxygen species and superoxide anions [21,22,23]. It has also been proved to be an effective antimicrobial agent against a broad range of pathogenic microorganisms, including *E. faecalis* [24,25,26,27,28]. In the field of biomaterial and bioengineering, quercetin has also been widely utilized for biomodification of scaffold materials due to its glutaraldehyde-comparable collagen-crosslinking capability and superior cytocompatibility [29]. Recently, it has been doped into dentin adhesive with the aim to improve the long-term performance of the resin–dentin bonding interface, wherein quercetin protects the structural integrity of dentin collagen from biodegradation by crosslinking the apatite-depleted collagen [30]. The anti-biofilm and collagen-stabilizing effects of quercetin make it a potential candidate for an endodontic irrigant.

The objective of the current study was to explore the feasibility of quercetin solution as an auxiliary endodontic irrigant with anti-biofilm activities against *E. faecalis* and a dentin collagen-stabilizing capability. We evaluated the anti-biofilm effect of quercetin by confocal laser scanning microscopy (CLSM), based on an in vitro dentin infection model with intensive and ubiquitous *E. faecalis* infection in the dentin tubules [31]. The dentin collagen-stabilizing capability of quercetin solution was assessed with regard to biomechanical properties and biodegradation resistance of dentin collagen. Moreover, biomodification of demineralized dentin collagen by quercetin was characterized by X-ray photoelectron spectroscopy (XPS), and the cytotoxicity of quercetin irrigant was tested on human dental pulp cells (HDPs). The null hypotheses are quercetin irrigant (1) does not kill *E. faecalis* or inhibit the growth of biofilm; (2) does not enhance dentin collagen’s modulus of elasticity and reduce the loss of modulus and hydroxyproline release; (3) shows cytotoxicity on HDPs.

## 2. Materials and Methods

### 2.1. Experimental Irrigant Preparation and Tooth Collection

Due to the poor solubility in water, we selected pure ethanol as the solvent for quercetin in our study [32]. Quercetin powder (Sigma–Aldrich, St. Louis, MO, USA) was directly added to 100% ethanol, followed by water-bath heating at 37 °C for 15 min. All experiments were conducted based on the following 5 groups: sterile water (control group), 0 wt% (ethanol group), 1 wt%, 2 wt% and 4 wt% quercetin groups.

All methodologies applied in this study were performed in accordance with the relevant guidelines and regulations [33]. Human single-rooted teeth (antibacterial test and XPS characterization) and third molars (biomechanical test) extracted for the orthodontic treatment were collected with informed consent. Different fresh and intact human teeth were obtained for the following purposes: (a) 10 single root premolars for antimicrobial test; (b) 40 dentin beams (1.2 × 0.8 × 6.0 mm^3^) prepared from the third molars for biomechanical test; and (c) 5 premolars for XPS examination. All teeth were stored in saline solution at 4 °C for no more than one month [34].

### 2.2. Antibacterial Test

#### 2.2.1. Sample Preparation

As previously described [35], each single-rooted tooth was horizontally sectioned 1 mm below the cementoenamel junction using a low-speed diamond saw (SYJ-150, Kejin, Shenyang, China) at 300 rpm under water cooling. Then, the root was cut in parallel with the occlusal surface of the teeth to obtain a cylindrical root dentin block with a length of 4 mm. The root canals inside the dentin block were enlarged to a diameter of 1.5 mm using a size 6 Gates Glidden drill (Tulsa Dentsply, Tulsa, OK, USA) at 300 rpm under water cooling. A groove was made in the middle of each cylindrical dentin block using a low-speed handpiece (Tulsa Dentsply, Tulsa, OK, USA) with a round bur. Then the dentin block was fractured with a blade and a hammer into 2 semi-cylindrical halves. The semi-cylindrical halves were ground to 2 mm thickness by 600-grit silicon paper. The specimens were then shaped using a low-speed handpiece with a fine carbide bur at 300 rpm under water cooling to obtain refined semi-cylindrical specimens with a size approximately 4 × 4 × 2 mm^3^, so that they can be placed in a 0.45 μm pore size filter tube (Pall Corporation, Ann Arbor, MI, USA) [31].

The specimens were rinsed with a 5.25% sodium hypochlorite and 6% citric acid (pH 4.0) solution in an ultrasonic bath (Sankei Giken Industry Co Ltd., MIE, Tokyo, Japan) for 4 min to remove the smear layer, and were rinsed in water for 1 min. Each specimen was placed in a filter tube with the canal side up. The gap between the specimen and the inner walls was sealed with resin composite (Kerr Co, Orange, CA, USA).

#### 2.2.2. Infection of Dentin

*E. faecalis* ATCC 29,212 (American Type Culture Collection, Manassas, VA, USA) was cultured on brain–heart infusion (BHI) agar (Becton–Dickinson, Sparks, MD, USA) plates at 37 °C overnight. Then, the bacteria were suspended in a BHI broth and standardized spectrophotometrically to 3 × 10^6^ colony-forming units (CFU)/mL (OD_405_ = 0.05) [31]. Then 500 μL of the bacterial suspension was added into each filter tube wherein the dentin specimen had been placed. The tubes were centrifuged at forces of 1400× *g*, 2000× *g*, 3600× *g*, and 5600× *g* in a sequence twice each for 5 min. After dumping the last centrifugal solution, the upper compartment was refilled with a fresh 500 µL of bacterial suspension before the next force of centrifugation. Ten filter tubes containing specimens were incubated in a sterile BHI broth in air at 37 °C for 1 week.

After incubation, the infected specimens were taken out from the filter tube and the resin composite was removed. The specimens were rinsed by sterile water for 1 min and air-dried. Then, nail varnish was used to seal the cemental sides of the dentin pieces. Subsequently, a total of 20 specimens were randomly distributed into 5 groups with different irrigants as previously described: control group, ethanol group, 1 wt%, 2 wt% and 4 wt% quercetin groups. A droplet (50 μL) of irrigant was placed on the pulp side of the dentin pieces for 3 min. Then, the specimens were rinsed with sterile water for 1 min and fractured vertically through the root canal into two halves to expose a fresh surface of longitudinally fractured dentinal tubules for CLSM examination.

#### 2.2.3. Confocal Laser Scanning Microscopy (CLSM) Examination

The fractured dentin specimens were stained with LIVE/DEAD BacLight Bacterial Viability Kit (Molecular Probes, Eugene, OR, USA) according to the manufacturer’s instructions. The excitation/emission wavelengths of SYTO 9 (green fluorescent dye that stains live bacteria) and propidium iodide (PI, red fluorescent dye that stains dead bacteria) were 480/500 nm and 490/634 nm, respectively. The mounted specimens were observed by CLSM (LSM 710, Zeiss, Oberkochen, Germany) using a 20× lens with an additional zoom of 2×. Under the microscope, the border of root canal for each specimen was first located, and then five randomly selected locations starting from the root canal border were scanned. For any scanned location, one image stack containing 20 slices were acquired at 0.5-μm z-step (the distance between two adjacent images of a stack). Three-dimensional reconstructions and volume calculation of all image stacks were performed using Imaris 7.2 software (Bitplan Inc., St Paul, MN, USA). The proportion of dead bacteria within the *E. faecalis* biofilms was represented by the volume ratio of red fluorescence to green and red fluorescence.

### 2.3. Biomechanical Test

#### 2.3.1. Modulus of Elasticity of Demineralized Dentin before and after Irrigation

Forty dentin beams (1.2 × 0.8 × 6.0 mm^3^) free of notches and cracks were sectioned from the root dentin of the third molars as previously described [36]. They were completely demineralized in 0.5 M ethylenediaminetetraacetic acid (EDTA) (pH = 7.5) for 7 days at 37 °C and were thoroughly rinsed with distilled water for 10 min. The demineralized beams were divided into 5 groups and immersed in 2-mL irrigants for 3 min, followed by water-rinsing for 1 min. A bacterial collagenase solution was prepared by dissolving bacterial collagenase (from Clostridium histolyticum, 125 collagen digestion units per mg solid; Sigma–Aldrich, Merck KGaA, Darmstadt, Germany) in a 0.4 M ammonium bicarbonate-buffered solution (pH = 7.9) at a concentration of 0.1% (m/m). The beams were incubated in the bacterial collagenase solution individually in a 96-well plate for 48 h at 37 °C. The modulus of the elasticity of all beams was estimated at baseline, after 3-min treatment in their irrigant solutions and after 48-h enzymatic hydrolysis through three-point bending flexure test. The cross-head of the universal tester (EZ Graph, Shimadzu, Kyoto, Japan) is used at a rate of 1 mm/min in a controlled range of up to 10% of stain [37]. The corresponding modulus of elasticity was calculated according to the formula using pressure–displacement curves.
E = mL^3^/4bT^3^ (MPa),(1)
where m: slope; L (mm): support arm spacing; b (mm): sample width; T (mm): sample thickness.

#### 2.3.2. Hydroxyproline (HYP) Release Assay

After the aforementioned 48-h collagenase incubation, the dry mass of each beam was measured and the HYP content in the collagenase incubation medium was analyzed by using a HYP kit (Jiancheng, Nanjing, China). The HYP content was used to estimate the percentage of degraded collagen, based on the knowledge that 90% of the dry mass of demineralized dentin consists of type I collagen and that dentin collagen contains 9.6% of hydroxyproline [38]. The absorbance was measured at 550 nm with a spectrophotometer (Tecan Group, Männedorf, Switzerland) and the HYP released from each beam was represented by dividing the HYP by the dry weight of dentin beam (μg HYP/mg dentin).

### 2.4. X-ray Photoelectron Spectroscopy Characterization

Demineralized dentin discs (4 × 4 × 1 mm^3^) were treated by different irrigants for 3 min each, and were rinsed with deionized water for 1 min. Then, the samples were dried for 12 h in a vacuum pump (FY-4C-N, Value, Hangzhou, China). X-ray Photoelectron Spectrometer (ESCALAB250Xi, Thermo Fisher Scientific, Waltham, MA, USA) with monochromatic Al Kα X-ray source was used to characterize the elemental distribution of C1s and O1s on the dried discs. Five sites of the fractured surfaces of each specimen were randomly selected for the analysis. The acquired data were analyzed using XPS PEAK 4.1 software for peaks of each element. The high resolution of C1s and O1s spectra was curve-fitted based on Gaussian–Lorentzian composite function with the background subtracted in Shirley mode.

### 2.5. Cytotoxicity Evaluation with CCK-8 Assay

Human dental pulp cells (HDPs) were a gift from the Department of Stomatology, Union Hospital, Huazhong University of Science and Technology. The cells were cultured in α-MEM (HyClone, Logan, UT, USA) and 100 μg/mL of penicillin G-streptomycin at 37 °C with 5% CO_2_ in a humidified environment. The third passage of HDPs were seeded into 96-well plates (5000 cells per well) for 24 h, followed by a 24-h treatment with 100-fold dilutions of the previous test solutions in α-MEM. Thereafter, the Cell Counting Kit-8 (MA0218-1, Meilunbio, Dalian, China) was performed according to the manufacturer’s protocol for 4 h to assess cell viability (%). The experiment was performed in quintuplicate. The experimental process was shown in Figure 2.

### 2.6. Statistical Analysis

Data are presented as the mean ± standard deviation. Statistical analysis was performed using the SPSS 25.0 software (SPSS Inc., Armonk, New York, NY, USA) package at the 5% level of significance. All data were submitted to one-way analysis of variance (ANOVA) and complemented by Tukey’s multiple comparison. Groups labeled with the same letters are not statistically different (*p* > 0.05). Power analysis was conducted to determine the sample size for each experiment and to project the power as 0.8 and significance level as 0.05.

## 3. Results

### 3.1. Effect of Quercetin on E. faecalis

The bacteria penetration from the root canal to dentin tubules after centrifugation and the anti-biofilm effect of each irrigant against *E. faecalis* are shown in Figure 3. The histogram represents the death cell rate of *E. faecalis* biofilm that grew in infected dentin tubules. There were statistically significant differences between the experimental and the control groups (*p* < 0.05). Both the pure ethanol and quercetin solutions showed obvious anti-biofilm effects, and as the concentration of the quercetin solution increased, the proportion of dead volumes of *E. faecalis* raised from 19.13% to 57.8%, with a statistically significant difference (*p* < 0.05). The 4 wt% group (57.8%) showed the strongest anti-biofilm activity (*p* < 0.05).

### 3.2. Effect of Quercetin on Dentin’s Biomechanics

Figure 4 shows that all quercetin groups had a significantly higher elastic modulus, and less modulus loss and HYP release than that of the control groups (*p* < 0.05). Demineralized dentin beams in the control and pure ethanol group had no significant change of modulus (*p* > 0.05) and had the most reduction in elastic modulus after enzymatic degradation (*p* < 0.05). However, dentin beams in the quercetin groups, in concentration between 1 to 4 wt%, had a higher increase in elastic modulus after irrigant treatment (*p* < 0.05), and the lower loss of elastic modulus reduced after biodegradation (*p* < 0.05), ranging from 55.5% to 21.93%. After 48-h collagenase incubation, dentin beams from the control group released 520.9 ± 115.6 μg of HYP/mg dentin, whilst beams irrigated by quercetin demonstrated a lower HYP release (*p* < 0.05), ranging from 88.15 ± 37.6 to 296.9 ± 45.6 μg of HYP/mg dentin. The improvement of the elasticity and biodegradation resistance of dentin was quercetin concentration-dependent (*p* < 0.05).

### 3.3. Characterization of Crosslinking between Quercetin and Collagen

The XPS analysis characterized the chemical bonds of dentin collagen after quercetin biomodification as a surface-specific technique (Figure 5). The C1s spectra were curve-fitted into three main peaks at 284.7, 285.9 and 288.0 eV, which are associated with carbons in the aliphatic chain of amino acid side chains (C-C/C-H), carbons at the α-position of peptidic residues (C-O/C-N) and carbons in peptidic carbonyl and in carboxyl (C=O), respectively [39]. The O1s spectra exhibits two main peaks at 531.0 and 532.2 eV, which are dominantly attributed to the ketonic oxygen (C=O) and hydroxyl oxygen (C-O), respectively [40]. The detailed parameters of each peak of C1s and O1s spectra were listed in Table 1. Due to the abundance of hydroxyl groups in quercetin molecules, the C-O peak areas of both C1s and O1s spectra increased, in a quercetin concentration-dependent manner. The relative area of the C-O peak under C1s from in the quercetin groups increased from 13.0% to 46.4%, and the area of the C-O peak under O1s increased from 26.2% to 68.8%, which indicated that quercetin with abundant phenolic hydroxyls was grafted onto dentin collagen.

### 3.4. Effect of Quercetin on HDPs Viability

Using human dental pulp cells, we investigated the cytotoxicity of quercetin solution (Figure 6). Following a 24-h exposure, the quercetin irrigants presented acceptable cytotoxicity compared with the control group (*p* > 0.05), even at the highest concentration tested.

## 4. Discussion

An essential step in successful root canal treatment is to minimize the number of microorganisms in the root canal and dentin tubules through mechanical instrumentation and the use of irrigants with antimicrobial properties. However, due to the complexity of the root canal system (e.g., isthmus, lateral canals, root canal side, traffic branch, and root separation), it is difficult to remove all bacteria in the dentin tubules by routine mechanical methods or endodontic irrigants [41,42]. In addition, lurking in the root canal system, bacteria can withstand harsh environmental conditions and exhibits resistance to many antibiotics [4,43]. *E. faecalis* is physically and ecologically strong and can easily penetrate deep into dentin tubules [44,45]. Although several endodontic irrigants have been investigated, some compounds demonstrate many adverse effects such as compromised efficiency, cytotoxicity to mammal cells, allergic potential and even reduction in the mechanical properties of dentins [46,47,48]. Some studies have confirmed that chlorhexidine solution cannot infiltrate into dentinal tubules wherein a great number of infectious bacteria could hide for survival but can decrease the microhardness of dentin [11,42,49]. Sodium hypochlorite can penetrate the dentin and degrade the collagen matrix, eventually leading to root fracture [50,51]. Given this, the search for an adjunctive endodontic irrigant with strong antibacterial properties, the dentin collagen-stabilizing ability and good biocompatibility is paramount. In previous studies, quercetin has been proven to promote the long-time resin–dentin bonding [30,52]. In this study, we evaluated the utility of quercetin as auxiliary endodontic disinfectant and its effect on mechanical properties and degradation resistance of dentin, and rejected the proposed hypotheses according to the results.

In order to eliminate the effect of ethanol on the experimental results, 100% ethanol was utilized as the negative control since it was used to dissolve quercetin. Ethanol has a strong permeability and can infiltrate into the inner parts of bacteria and fungi to denature proteins crucial for their survival. Previous studies have proved that 95% ethanol had antibacterial effects against mono- and multispecies biofilms as an endodontic irrigant [53]. In addition, ethanol can improve the sealing ability of the root canal obturation decrease the leakage as a final irrigant, and has no effect on endodontic irrigants’ antimicrobial properties [54,55,56]. Our experiments also show that ethanol solutions at appropriate concentrations are not significantly cytotoxic to HDPs, indicating that ethanol is a suitable solvent for our study.

By establishing an in vitro noninvasive model of infected root dentin tubules, we for the first time demonstrated the antibacterial ability of quercetin-containing ethanol solutions against *E. faecalis* biofilms hidden in the dentin tubules (Figure 3). With increased concentration, the antibacterial effect of the quercetin solution was enhanced with deeper penetration into the dentin tubules (*p* < 0.05). Quercetin is a natural polyphenolic flavonoid that presents in many kinds of plants. It has a strong antimicrobial ability against various Gram-positive bacteria, Gram-negative bacteria, and viruses [26,57]. It has been reported that quercetin can change the membrane potential to hinder bacteria’s ability to synthesize adenosine triphosphate (ATP) and transporting material, thereby killing the bacteria [58]. However, it is hard to guarantee the antimicrobial ability in infected dentin tubes in vivo. Before the development of this noninvasive model, dentin block culture had been used as a very common method to simulate the presence, action, and eradication of bacteria in dentin tubules. However, it is difficult or impossible to obtain substantial, evenly distributed bacterial presence in dentin tubules with this method [59,60]. The application of this new model allows predictable, intensive and deep penetration of bacteria in dentin tubules, greatly aiding the study of dentin disinfection. In combination with the CLSM test, we not only directly visualized the bacteria, but also identified live and dead bacteria in infected dentins, which verified this in vitro noninvasive model as a suitable detection method with clinical relevance [7,61].

Our results showed that quercetin irrigant increased the biomechanical properties and biodegradation resistance of dentin collagen (Figure 4A,B). Because of the abundance of phenolic hydroxyl groups on its molecules, quercetin may interact with the collagen in root dentin via hydrogen bonds, van der Waals forces, electrostatic forces, and hydrophobic forces, thus maintaining the biostability of the demineralized collagen matrix after irrigant application [62], and improving the mechanical properties of demineralized dentin [63]. In the current work, we utilized XPS to confirm the quercetin-biomodificaiton of dentin collagen through the increased C-O peak area under the C1s and O1s spectra (Figure 5 and Table 1) [39,64]. We also confirmed the resistance to collagenase degradation of quercetin-treated dentin by HYP release assay. Dentin collagen treated by high concentrations (i.e., 2 wt%, 4 wt%) of quercetin showed the least HYP release (Figure 4C). Flavonoids, such as quercetin, inhibit both free and collagen-bound proteolytic enzymes in dentin [65] and down-regulate endogenous protease expression [66] to inactivate the protease, preventing free access of collagenase to sites containing collagen chains, thus increasing collagen’s resistance to enzymatic degradation [67,68]. In addition, quercetin crosslinks with the exposed dentin to provide a mechanical barrier to the collagen matrix, with protection of the remaining dentin against acid attack erosion [69], abrasion [70] and erosion plus abrasion [71].

Since endodontic irrigants work directly in contact with human tissues in the clinic, they should be biocompatible to human cells and tissues. Quercetin is known to have good biosafety and low cytotoxicity [72]. According to the CCK-8 results (Figure 6), quercetin-containing ethanol solution with low concentration had no cytotoxic effect on human dental pulp cells. However, the other results argued the cytotoxic of quercetin with high concentration [73]. However, a previous study has emphasized that irrigants generally do not flow all the way to the root end in the positive pressure of irrigation technique [74]. Therefore, it is proper to use quercetin as an auxiliary endodontic irrigant under a proper isolation condition and technical conditions.

We believe quercetin has great potential for dental applications due to its multiple properties. For example, the success of the regenerative endodontic procedures (REP) also depends on increasing the thickness and length of the root canal wall and eliminating the infection in the root canal and apical area, which may indicate that quercetin as an auxiliary root canal irrigants has the potential application for REP [75]. However, there are two issues that need to be considered in our study. The results of antimicrobial tests and 48-h collagenase degradation study are not sufficient enough to draw conclusions about the long-term antimicrobial activity. Therefore, further studies are needed to assess the durability of dentin bio-modification and the antimicrobial activity of quercetin solutions. The second issue is that high concentrations of polyphenolic flavonoids, such as antioxidants, may affect the polymerisation reaction of resin root canal sealers in root canal treatment procedures [76,77]; further researches are needed to explore the reaction with endodontic materials.

## 5. Conclusions

Quercetin-containing ethanol irrigants showed a bactericidal effect against *E. feacalis* biofilms in dentin tubules and can maintain the mechanical strength of demineralized dentin and enhance biodegradation resistance. This proof-of-concept study demonstrated the utility of quercetin as a safe and reliable adjunctive root canal irrigant. Further clinical studies are still needed to evaluate its effectiveness in combination with other root canal disfectants, as well as the biosafety in vivo.

## Figures and Tables

**Figure 1 materials-14-01178-f001:**
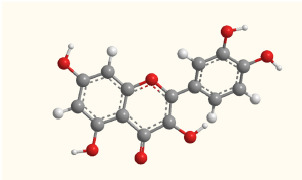
Quercetin’s chemical structure (molecular formula: C_15_H_10_O_7_).

**Figure 2 materials-14-01178-f002:**
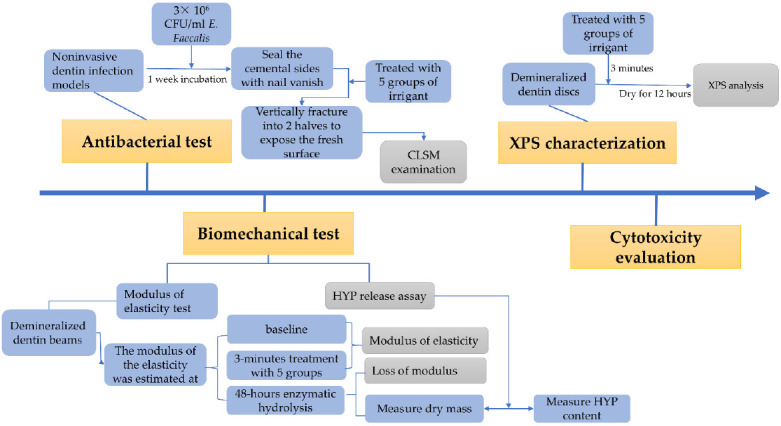
The flow chart for the experimental process. CFU: colony-forming units; HYP: hydroxyproline; XPS: X-ray photoelectron spectroscopy; CLSM: confocal laser scanning microscopy.

**Figure 3 materials-14-01178-f003:**
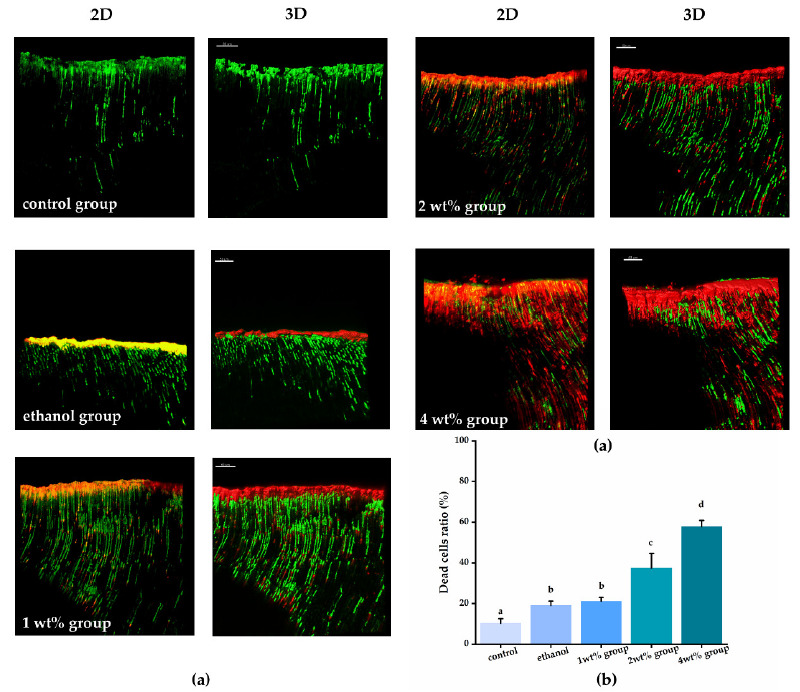
(**a**) Representative CLSM images of *E. faecalis* biofilms in dentinal tubules treated by different irrigants after live/dead staining and histogram showing dead cell ratio of bacteria biofilms from corresponding groups. For the CLSM images from each group, the image on the left is a 2-dimensional emerged image and the image on the right is a 3-dimensional reconstruction. (**b**) Quantitation data of the percentage of live and dead area, groups with the same letters are not statistically different (*p* > 0.05).

**Figure 4 materials-14-01178-f004:**
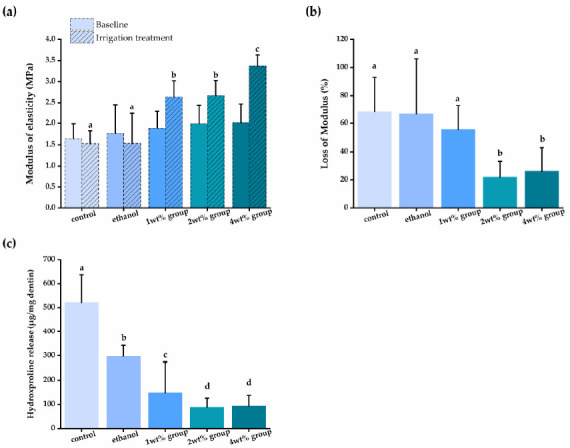
(**a**) The elastic modulus (MPa) of demineralized dentin beams at baseline, after irrigant treatment; (**b**) the loss of modulus (%) of demineralized dentin beams after collagenase degradation compared to dentin beams treated by different irrigants; (**c**) the released HYP (μg/mg dentin) from dentin treated by different irrigants and challenged by collagenase. Groups with the same letters are not statistically different (*p* > 0.05).

**Figure 5 materials-14-01178-f005:**
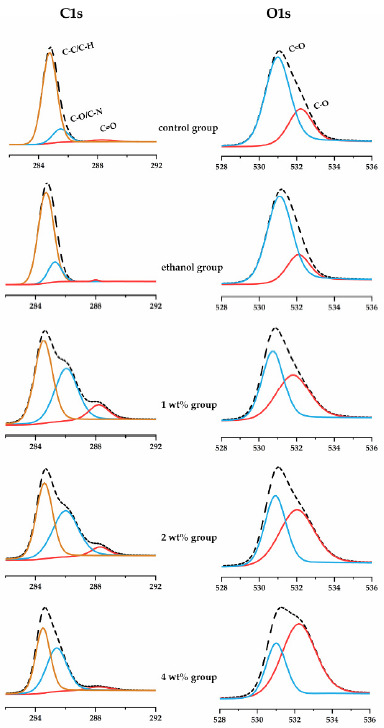
XPS high-resolution C1s (left column) and O1s (right column) spectra of dentin treated by irrigants.

**Figure 6 materials-14-01178-f006:**
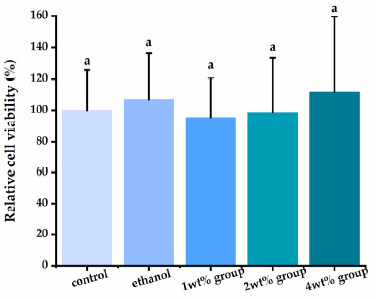
Evaluation of human dental pulp cells’ (HDPs’) viability exposed to different quercetin-containing mediums. Groups with the same letters are not statistically different (*p* > 0.05).

**Table 1 materials-14-01178-t001:** Data of C1s and O1s XPS narrow-scan spectra for dentin treated by irrigants.

	C1s			O1s	
	C-C/C-H	C-O	C=O	C=O	C-O
**Control group**					
BE ^a^ (eV)	284.7	285.5	288.3	530.9	532.2
FWHM ^b^ (eV)	1.2	1.2	1.7	1.6	1.5
Area (%)	84.2	13	2.8	73.8	26.2
**Ethanol group**					
BE (eV)	284.7	285.3	288	531.1	532.1
FWHM (eV)	1.2	1.1	1	1.6	1.4
Area (%)	83.6	15.9	0.5	79.1	21.9
**1 wt% group**					
BE (eV)	284.5	286	288.2	530.7	531.9
FWHM (eV)	1.4	1.8	1.5	1.4	2
Area (%)	49.7	40.6	9.7	55.9	44.1
**2 wt% group**					
BE (eV)	284.6	286	288.3	530.9	532.2
FWHM (eV)	1.3	2	1.4	1.4	2
Area (%)	48	46	6	54.2	45.8
**4 wt% group**					
BE (eV)	284.5	285.4	288.2	531	532.2
FWHM (eV)	1.1	1.6	1.8	1.3	2.1
Area (%)	49.4	46.4	4.2	31.2	68.8

^a^ BE: binding energy; ^b^ FWHM: full width at half maxima.

## Data Availability

Not applicable.

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
