# Peer review of "Quercetin as an Auxiliary Endodontic Irrigant for Root Canal Treatment: Anti-Biofilm and Dentin Collagen-Stabilizing Effects In Vitro"

_materials, 2021, doi:10.3390/ma14051178_

Round 1

Reviewer 2 Report

Nice paper. The study is well conducted and argumented, but this is a preclinical study. The treatment of the dentin is so far from the real clinical condition of a root canal therapy.  I rather prefer you make clear to the readers the  nature of this study, adding "in vitro" to the title.

Line 16 "..that strengthen...". I rather suggest ..that doesn't weak....

Finally, please check some little flaws (like" irrgant"- line 160)

Reviewer 3 Report

INTRODUCTION

Please state the null hypotheses of the study at the end of the introduction

MATERIALS AND METHODS

Line 84-85: please add references for the guidelines and regulations

Line 91: please add a reference for the type of storage

Line 116: I think that bacteria should be read at OD600. If not, please add a reference for OD 405

Line 189 and 192: please write “α-MEM”

Line 197: please add the manufacturer, city and state for SPSS

RESULTS

Please divide results into sections, according to each experimental phase.

DISCUSSION

Please state whether statistical null hypothese were accepted or refused.

Lines 266-269: please add references for the adverse effects demonstrated by some compounds

At the end of the discussion, please add the limitations of the study

It should be interesting to discuss that further studies should consider should evaluate the effectiveness of quercitin considering other parameters, like temperature, time of exposure, etc.

Round 2

Reviewer 1 Report

This research is under the scope of this journal; the topic is interesting for readers and this research deals with potentially significant knowledge to the field and an open new way for future studies.

The authors improved the quality of the manuscript after the reviewer's indications. Congratulations!